# Pacifier Usage Among Saudi Children: A Cross-Sectional Study in Jeddah, Saudi Arabia

**DOI:** 10.3390/healthcare13151935

**Published:** 2025-08-07

**Authors:** Sara M. Bagher, Logain Alattas, Haneen Bakhaidar, Najat M. Farsi, Shahad N. Abudawood, Heba Jafar Sabbagh

**Affiliations:** 1Pediatric Dentistry Department, Faculty of Dentistry, King Abdulaziz University, P.O. Box 80209, Jeddah 21589, Saudi Arabia; sbagher@kau.edu.sa (S.M.B.); hbakhaidar@kau.edu.sa (H.B.); nfarsi@kau.edu.sa (N.M.F.); sabudawood@kau.edu.sa (S.N.A.); 2Tam Dental Clinic, Jeddah 23415, Saudi Arabia; logain.alattas@gmail.com

**Keywords:** breast feeding, childbirth order, maternal knowledge, pacifier, Saudi Arabia, nursery

## Abstract

**Background/Objectives:** Pacifier use in infants has both beneficial and harmful implications, and dipping pacifiers in sweeteners is used by some parents to soothe infants. This study aimed to assess pacifier usage among mothers in Jeddah, Saudi Arabia, and to examine its association with child demographics, maternal socioeconomic status (SES), and maternal knowledge of the risks associated with dipping pacifiers in sweeteners. **Methods:** A cross-sectional study was conducted among mothers of healthy children aged 2 to 4 years during community-awareness events in Jeddah. Participants completed a validated Arabic questionnaire covering pacifier use patterns, feeding practices, SES background, and knowledge regarding the adverse effects of pacifier sweetening. **Results:** A total of 1438 mothers participated. The mean age of children was 34.3 ± 10.7 months, with 441 children (30.7%) reported as pacifier users. Among them, 202 (45.8%) used pacifiers both during the day and at night. Most children (35.6%) discontinued use between 4 and 6 months of age. Only 6.1% of mothers reported adding sweeteners to pacifiers. Pacifier usage was significantly higher among first-born children (37.6%, *p* < 0.001), those who received both bottle- and breastfeeding (65.5%, *p* < 0.001), and children enrolled in nursery (62.1%, *p* = 0.007). Most mothers (75.6%) were aware of the link between sweetened pacifiers and dental caries, and 69.4% of those who had this knowledge avoided giving their child a pacifier (*p* = 0.077). **Conclusions:** Birth order, feeding practices, and nursery attendance are key predictors of pacifier use. Enhancing parental awareness and education may support early interventions to promote healthier oral and feeding habits in young children.

## 1. Introduction

Non-nutritive sucking habits (NNSHs), such as pacifier usage and thumb or finger sucking, are common in babies and young children and naturally decline as they grow older. For some, this habit may persist for several years, increasing the risk of malocclusion in the primary dentition, which may carry over into the permanent teeth [1]. Pacifiers are commonly used to calm and soothe infants and toddlers, helping them fall asleep and reducing fussiness [2]. Pacifier usage can provide protection against sudden infant death syndrome, with studies suggesting up to 90% risk reduction, particularly when used during sleep [3,4]. While pacifiers have some advantages, there are several potential disadvantages to their prolonged and extensive use. It can lead to dental issues and malocclusion, such as increased overjet, anterior open bite, and/or a posterior crossbite [5]. The severity of the resulting malocclusion is influenced by the duration, frequency, and intensity of use [6]. Furthermore, dental malocclusion resulting from NNSHs can indirectly compromise periodontal health. Some reports consider malocclusion as a predisposing factor for dental caries and periodontal problems since tooth misalignment causes accumulation of bacterial plaque and hinders its removal and proper oral hygiene practices [7,8]. In addition, studies report that pacifier use may increase candida and lactobacilli, elevating the risk of oral infections and gingival inflammation [9,10]. Therefore, pacifiers should be disinfected regularly.

Moreover, parents play important roles in establishing dental care at home and maintaining oral hygiene for their young children. Pediatric dentists should counsel and provide anticipatory guidance regarding oral hygiene practices, including brushing twice a day, using an age-appropriate toothbrush and fluoridated toothpaste, along with the safety, benefits, and risks of using pacifiers to parents of infants and toddlers who choose to use them [11]. Multiple global organizations such as the American Academy of Pediatric Dentistry (AAPD) and American Academy of Family Physicians align in their recommendations and [12] support the parental decision to introduce the pacifier based on their preference and their child’s needs, but at the same time, they acknowledge the potential disadvantages of prolonged or excessive pacifier usage and provide guidelines on its use, emphasizing risks such as dental problems, effect on jaw growth and oral muscles [13], ear infections, reduction in breastfeeding duration [14], and speech development issues [15]. The recommendation further goes on to recommend weaning their children off pacifiers by 36 months to prevent the risk of long-term dependency and potential negative effects. Scientific evidence suggests that dental changes and malocclusion can be reversible and self-corrected if the pacifier usage is discontinued between two and three years [16].

A cross-sectional study in Saudi Arabia reported that most of the mothers (68.3%) had the knowledge that using a pacifier could affect the primary teeth and cause an open bite [17]. In addition, a recent study in Turkey aimed to evaluate maternal knowledge and behavior toward the use of pacifiers, and 35.5% of them reported that using a pacifier is harmful and can interfere with teeth and jaw development [18].

The practice of dipping pacifiers in sweeteners, whether natural or artificial, such as honey or sugar, is traditionally used by some parents to soothe infants. However, this practice increases the risk of dental caries by promoting the growth of harmful bacteria in the mouth, leading to dental caries [10,19]. Mothers’ and caregivers’ knowledge and awareness regarding the potential risks of using pacifiers are important to prevent harm to their children. Moreover, there has been limited research on pacifier usage in Saudi Arabia. Therefore, this study aims to assess pacifier usage among a group of Saudi 2- to 4-year-old healthy children in Jeddah and to identify the relationship between pacifier usage and the child’s demographic data (age, gender, birth order, and type of delivery), the child’s feeding practices, attending a nursery, maternal socioeconomic status (education and occupation), average family monthly household income, and knowledge of the harmful effect of dipping the pacifier in sweeteners.

## 2. Materials and Methods

This cross-sectional study was conducted in the city of Jeddah in Saudi Arabia after obtaining ethical approval from the ethics committee of King Abdulaziz University (131-11-18 on 23 January 2019).

Data for this study were collected between September and December 2019. The inclusion criteria included Saudi mothers of healthy 2- to 4-year-old children living in Jeddah, Saudi Arabia. The recruitment of the participants took place during three different community-awareness activities organized by the King Abdulaziz University Faculty of Dentistry (KAUFD) across the central, northern, and southern areas of Jeddah city.

Eligible mothers were approached by one of the two trained dentists on the research team to introduce them to the research aim. Those who agreed to participate signed an Arabic consent form before being interviewed using a validated Arabic questionnaire. If a mother had more than one healthy child within the age range, she was asked to answer the questionnaire for her youngest child.

Stratified sampling was used to select the three community-awareness activities in Jeddah’s central, northern, and southern regions. For sample size calculation, OpenEpi, Version 3, was used with a power of 80%, a two-sided confidence level (1-alpha) of 95%, and an odds ratio of 1.52, as referenced by Carcavalli et al. (2018) [20]. This estimated a sample size of 1188 participants.

The questionnaire was pre-tested for both face and content validity. Ten representative subjects were interviewed and asked for their opinions on the questions’ feasibility for face validity. Based on their feedback, minor changes were made to some of the questions.

The questionnaire was divided into three parts and included a total of 13 close-ended questions. The first part included the child’s demographic data (age, gender, birth order, and type of delivery), the child’s feeding practices (exclusive breastfeeding, exclusive bottle feeding, and mixed feeding), attending nursery, and maternal socioeconomic status (SES) (education and occupation), and average family monthly household income. The average family monthly household income was categorized as low (below 7000 Saudi Riyals), middle (7000 to 16,000 SR), or high (above 16,000 SR) based on the statistics provided by Saudi Arabia’s official website for family income data [21].

In the second part of the questionnaire, the pacifier usage included questions regarding the time the child used the pacifier during the day, the withdrawal age, whether the mother added sweeteners to the pacifier, and when solid food was added to the child’s diet.

At the end of the questionnaire, the mothers were asked to answer a knowledge question regarding the potential risk of dipping the pacifier in sweeteners, which can increase the risk of developing dental caries, followed by “yes,” “no,” or “I don’t know” answers. 

### Statistical Analysis

The data were analyzed using the Statistical Package for the Social Sciences, version 22.0 (SPSS Inc., Chicago, IL, USA). Frequency, percentages, and chi-square were calculated for categorical values. Binary regression was conducted for the relationship between pacifier usage (dependent factor) and gender, birth order, and type of delivery, the child’s feeding practices, and the maternal SES (independent factors) to overcome the effect of confounders. Adjusted odds ratios (AORs) and 95% confidence intervals (CIs) were calculated. The predetermined threshold of significance was established at a *p*-value of less than 0.05.

## 3. Results

### Demographic Characteristics

This study recruited 1438 mothers of children 4 years old or less. The mean age of participating children was 34.33 ± 10.7 months; 742 (51.6%) were females, and 441 (30.7%) used pacifiers. The demographic characteristics of the participating children and their mothers are presented in Table 1.

Of those who used pacifiers, 202 (45.8%) used it during daytime and nighttime. Most participating children, 157 (35.6%), stopped using the pacifier between the ages of four and six months, and only 27 (6.1%) mothers reported adding sweeteners to the pacifier. The characteristics of pacifier usage among the participating children are presented in Table 2.

Table 3 represents the association between the child’s demographic data and feeding practices, history of going to a nursery, and maternal SES with pacifier usage among children. The child’s birth order and type of feeding practices showed a highly significant association with using a pacifier (*p* < 0.001). The first-born children, 166 (37.6%), and children who received mixed feeding, 289 (65.5%), had the highest pacifier usage. In addition, attending a nursery showed a statistically significant relationship with the use of pacifiers (*p* = 0.007); children who are enrolled in a nursery were more likely to use of pacifiers, 167 (37.9%), compared to those who are not using a pacifier, 305 (30.6%). The child’s gender, type of delivery, and maternal SES did not show a statistically significant association with pacifier usage.

The regression analysis for the relationship between pacifier usage (dependent variable) and independent factors (child’s gender, birth order, feeding practices, attending a nursery) is shown. Child order showed a significant association with pacifier usage (*p* < 0.001). First-born children were statistically significantly more likely to use pacifiers (AOR: 2.351 and 95% CI: 1.703–3.246) than fourth-born or later. Moreover, breastfed children were significantly less likely to use a pacifier (AOR: 0.290 and 95% CI: 0.206–0.409) than those who received mixed feeding. Finally, children who attended a nursery were significantly more likely to use pacifiers (AOR: 1.325 and 95% CI: 1.035–1.695). The analysis did not include other factors that showed high collinearity with the included risk factors (Table 4).

Finally, this study examined the relationship between pacifier usage and maternal knowledge regarding the potential harm of dipping pacifiers in sweeteners. Most of the participating mothers, 1078 (75.6%), knew that dipping pacifiers in sweeteners can lead to dental caries. In addition, most of the participating mothers who knew the harmful effect of dipping the pacifiers in sweeteners did not give their participating child a pacifier, 748 (69.4%) (*p* = 0.077). The distribution of the participating mothers according to pacifier usage and their knowledge of the potential harm of dipping pacifiers in sweeteners is presented in Table 5.

## 4. Discussion

This study aims to assess pacifier usage among a group of Saudi 2- to 4-year-old healthy children in Jeddah and to identify the relationship between pacifier usage and the child’s demographic data (age, gender, birth order, and type of delivery), the child’s feeding practices, attending a nursery, maternal socioeconomic status (education and occupation), average family monthly household income, and knowledge of the harmful effect of dipping the pacifier in sweeteners.

The mothers of children aged 2 to 4 years were targeted in this study because this age group is more likely to still be using a pacifier and engaging in early feeding practices, making them highly relevant to our research objectives. Additionally, since the questionnaire relied on maternal recall, selecting a younger age group helped minimize recall bias and increased the likelihood of obtaining accurate and reliable information about recent or ongoing pacifier use and feeding behaviors.

Among our sample, only 441 (30.7%) used pacifiers, which is lower than the prevalence reported in Western countries [5,22]. Previous research has indicated that 79% of first-time Australian mothers introduced pacifiers to their infants [2]. Furthermore, a 2023 study conducted in Clark County, Nevada, revealed that 60.5% of participants provided pacifiers to their children [23]. The difference in results could be due to the cultural and religious background that supports and encourages breastfeeding for up to two years.

Of those who used pacifiers, 202 (45.8%) relied on them during both daytime and nighttime, a pattern consistent with findings from previous research by Saniatan et al. in 2023 [23]. This suggests that nearly half of the pacifier users in our study rely on them throughout the day rather than just at specific times, which may have a detrimental effect on primary dentition [15]. In addition, the pacifier withdrawal age varies among the participating children in our study, with a higher percentage of mothers, 157 (35.6%), reporting that their infants stopped using the pacifier between the ages of four and six months, which is consistent with another study conducted in Riyadh, Saudi Arabia, by Kakti et al., in 2019 [24], which is earlier than the recommended age by the AAPD. The AAPD advises discontinuing NNSHs, including pacifier usage, by 36 months. However, habits that persist beyond 18 months increase the risk of developing dental issues such as open bite, posterior crossbite, and class II malocclusion [15].

The practice of dipping pacifiers in sweetener substances like honey, juice, sugar, or syrup may be a common cultural practice in some regions of the world, as it may calm the child further and encourage them to suck more and accept the pacifier faster. In this study, only 27 mothers (6.1%) reported dipping the pacifier in sweeteners, a rate lower than that reported in other studies [18,25]. A Texas study surveyed parents of children up to 12 months old and found that 11% had given their infants honey pacifiers [25]. Furthermore, a study conducted in Turkey reported that 30.6% of the participating mothers engaged in the behavior of dipping pacifiers into sugary products, such as honey, molasses, and jam [18]. It is important to note that infants under 12 months should not be given honey as it may contain Clostridium botulinum spores, which can lead to infant botulism. This rare but serious condition affects the nervous system and can cause muscle weakness, breathing difficulties, paralysis, and even death [26].

In this study, the child’s birth order, type of feeding practices, and children who attend a nursery were associated with a higher chance of pacifier usage. First-born children were more likely to be given pacifiers than later-born siblings. This can be the case since first-time mothers usually lack previous experience and may give their children a pacifier more often. Also, first-born children usually receive more concentrated and focused parental care, increasing pacifier use. A study among first-time Australian mothers reported that 79% of them introduced pacifiers to their infants, and the primary reasons cited for the pacifier usage included soothing the baby and aiding sleep. Their decision may have been influenced by recommendations from family members, friends, or healthcare professionals [2]. In contrast, another study reported that mothers with more than one child were more likely to introduce a pacifier [23].

Moreover, most of the children who used pacifiers were mixed fed (289, 65.5%), while the exclusively breastfed children were significantly less likely (47, 10.7%) to use a pacifier. This finding aligns with a study by Gomes-Filho et al. (2019), who reported that exclusive breastfeeding is associated with a lower prevalence of pacifier use in children at 12 months [27]. Therefore, encouraging exclusive breastfeeding may help decrease pacifier usage and its potential adverse effects. However, the AAP recommends delaying the introduction of pacifiers until breastfeeding is established [28]. Moreover, a recent study has indicated that the timing of pacifier introduction, whether early or late, does not influence breastfeeding at six months [29].

The infant’s or toddler’s sex, type of delivery, and maternal SES did not show a statistically significant association with pacifier usage. This finding is consistent with previous research, which also reported no significant difference in pacifier usage based on the child’s sex [23]. In contrast, another study found a significant difference in pacifier usage between babies based on the delivery method. The highest prevalence of pacifier usage was seen among infants born via emergency cesarean section, while the highest rates of non-users were observed in vaginally delivered infants [30].

Furthermore, the study found no statistically significant association between maternal SES and pacifier usage. This contradicts Saniatan et al. and Mitev et al., who found that mothers who provide pacifiers to their children tend to be less educated compared to those who do not, indicating that SES may influence decisions regarding pacifier usage [23,31].

In addition, though most of the participating mothers, 1078 (75.6%), were aware that dipping pacifiers in sweeteners could have harmful effects, only 69.4% actively avoided giving a pacifier to their child. This suggests a critical knowledge–behavior gap. The disconnect indicates that awareness alone may not be sufficient to influence maternal practices, a phenomenon previously documented in health behavior research [32]. This could be attributed to habitual behaviors and cultural norms that override knowledge, such as the common practice of using pacifiers as soothing tools [33]. Additionally, the perceived risk may be underestimated; mothers might acknowledge the general danger and risk of sweetened pacifiers but fail to perceive the harm as immediate or serious enough to change their habits [34]. This finding highlights the need for targeted health education initiatives that go beyond raising awareness to effectively promote sustained behavioral change.

This study has provided valuable insights into the practices and maternal knowledge regarding pacifier usage in Saudi Arabia. One of the main strengths of this study is its large sample size, which represents the central, northern, and southern areas of the city of Jeddah. However, since the data were collected during community-awareness events, there is a risk of selection bias, and the sample may not accurately represent the general population. As a result, the generalizability of the findings is limited and may not fully reflect the broader national context. Furthermore, since this study relies on a questionnaire and self-reported data, there is a potential risk of recall bias. Participants may inaccurately recall or misreport past behaviors, such as pacifier usage, duration, or weaning age.

## 5. Conclusions

This study suggests that childbirth order, type of feeding practice, and attending a nursery are important predictors of pacifier usage among children in Saudi Arabia. Understanding these factors can help guide parental education and early intervention strategies to promote optimal oral health and feeding habits in children.

## Figures and Tables

**Table 1 healthcare-13-01935-t001:** The demographic characteristics of the participating children and their mothers (N = 1438).

Variables		N (%)
Child gender	Male	696 (48.4)
	Female	742 (51.6)
Birth order	1st	388 (27.0)
	2nd	401 (27.9)
	3rd	279 (19.4)
	4th or more	370 (25.7)
Type of delivery	Normal	963 (67.0)
	Cesarean	475 (33.0)
Type of feeding	Exclusive breastfeeding	345 (24.0)
	Exclusive bottle feeding	251 (17.5)
	Mixed feeding	842 (58.6)
Attending a nursery	Yes	472 (32.8)
	No	966 (67.2)
Maternal education	Illiterate/primary school	510 (35.5)
	Intermediate/high school	346 (24.1)
	College and postgrad	582 (40.5)
Maternal occupation	Student	110 (7.6)
	Worker	383 (26.6)
	Housewife	945 (65.7)
Average family monthly household income	Low	334 (23.2)
	Moderate	614 (42.7)
	High	490 (34.1)

**Table 2 healthcare-13-01935-t002:** Characteristics of pacifier usage among the participating children (N = 441).

Variables		N (%)
At what time of the day did the child use the pacifier?	Daytime	134 (30.4)
	Nighttime	105 (23.8)
	During both	202 (45.8)
Pacifier withdrawal age (in months)	0–3	54 (12.2)
	4–6	157 (35.6)
	7–12	52 (11.8)
	13–24	118 (26.8)
	25–36	49 (11.1)
	>36 m	11 (2.5)
Added sweeteners to the pacifier	Yes	27 (6.1)
	No	414 (93.8)
When was solid food introduced to the child’s diet (in months)?	0–3	10 (2.3)
	4–6	281 (63.7)
	7–12	117 (26.5)
	13–24	23 (5.2)
	24–36	10 (2.3)

**Table 3 healthcare-13-01935-t003:** Child demographic data and feeding practices, history of attending a nursery, maternal socioeconomic status, and the average family monthly household income with pacifier usage among the participating children (N = 1438).

Variables		Did the Child Use a Pacifier?		Total	*p*-Value
		Yes	No		
Child gender	Male	209 (47.4)	487 (48.8)	696 (48.4)	0.61
	Female	232 (52.6)	510 (51.2)	742 (51.6)	
Birth order	1st	166 (37.6)	222 (22.3)	388 (27.0)	<0.001 *
	2nd	114 (25.9)	287 (28.8)	401 (27.9)	
	3rd	72 (16.3)	207 (20.8)	279 (19.4)	
	4th or more	89 (20.2)	281 (28.2)	370 (25.7)	
Type of delivery	Normal	282 (63.9)	681 (68.3)	963 (67.0)	0.105
	Cesarean	159 (36.1)	316 (31.7)	475 (33.0)	
Type of feeding practice	Exclusive breastfeeding	47 (10.7)	298 (29.9)	345 (24.0)	<0.001 *
	Exclusive bottle feeding	105 (23.8)	146 (14.6)	251 (17.5)	
	Mixed feeding	289 (65.5)	553 (55.5)	842 (58.6)	
Attending a nursery	Yes	167 (37.9)	305 (30.6)	472 (32.8)	0.007 *
	No	274 (62.1)	692 (69.4)	966 (67.2)	
Maternal education	Illiterate/primary school	166 (37.6)	344 (34.5)	510 (35.5)	0.483
	Intermediate/high school	100 (22.7)	246 (24.7)	346 (24.1)	
	College and postgrad	175 (39.7)	407 (40.8)	582 (40.5)	
Maternal occupation	Student	42 (9.5)	68 (6.8)	110 (7.6)	0.152
	Worker	121 (27.4)	262 (26.3)	383 (26.6)	
	Housewife	278 (63.0)	667 (66.9)	945 (65.7)	
Average family monthly household income	Low	96 (21.8)	238 (23.9)	334 (23.2)	0.637
	Moderate	189 (42.9)	425 (42.6)	614 (42.7)	
	High	156 (35.4)	334 (33.5)	490 (34.1)	

* statistically significant at 0.05.

**Table 4 healthcare-13-01935-t004:** Regression analysis for the relationship between pacifier usage (dependent factor) and child gender, birth order, type of feeding practices, and attending a nursery (independent factors).

		AOR (95% CI) *p*-Value
Gender	Male	0.946 (0.748–1.196) 0.642
	Female	1.000
Birth order	1st	2.351 (1.703–3.246) < 0.001 *
	2nd	1.301 (0.935–1.811) 0.119
	3rd	1.185 (0.820–1.711) 0.366
	4th or more	1.000
Type of feeding practice	Exclusive breastfeeding	0.290 (0.206–0.409) < 0.001 *
	Exclusive bottle feeding	1.289 (0.957–1.736) 0.095
	Mixed feeding	1.000
Attending a nursery	Yes	1.325 (1.035–1.695) 0.026 *
	No	1.00

* statistically significant at 0.05.

**Table 5 healthcare-13-01935-t005:** Distribution of the participating mothers according to pacifier usage and their knowledge of the potential harm of dipping pacifiers in sweeteners (N = 1438).

Variables	Did the Child Use a Pacifier?		Total
Dipping the Pacifier in Sweetener Can Lead to Caries	Yes	No	
Yes	339 (76.8)	748 (68.8)	1078 (75.6)
No	22 (5.0)	31 (3.1)	53 (3.7)
I don’t know	80 (18.1)	218 (21.9)	298 (20.7)

*p* = 0.077.

## Data Availability

The raw data supporting the conclusions of this article will be made available by the authors on request.

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
