# Peer review of "Pacifier Usage Among Saudi Children: A Cross-Sectional Study in Jeddah, Saudi Arabia"

_healthcare, 2025, doi:10.3390/healthcare13151935_

Round 1

Reviewer 1 Report

Comments and Suggestions for Authors

This study presents an interesting and relevant exploration of pacifier use, particularly in the context of its potential association with early childhood caries. The authors took a valuable approach by assessing not only the prevalence of pacifier use but also maternal awareness of its associated risks, especially the practice of dipping pacifiers in sweeteners. Several points, however, warrant attention to strengthen the interpretation and implications of the findings:

  1. Data were collected during community-awareness events, the sample may not represent the general population, potentially introducing selection bias. This limitation should be clearly acknowledged.
  2. 75.6% of mothers were aware of the risks associated with sweetened pacifiers, but only 69.4% reported avoiding pacifier use, with the association not showing any statistical significance (p = 0.077). This indicates a gap between knowledge and behavior that need showing more variable may that related with this finding.
  3. the abstract structure is not allign with journal style.

Reviewer 2 Report

Comments and Suggestions for Authors

The authors explored pacifier usage among Saudi children using a cross-sectional study design. The manuscript seems very well-written and sound. The topic is interesting and literature in fact lacks regarding this issue, so that the information obtained in this study adds important data in the body of evidence on this topic. I have minor recommendations, as follows.

- The Introduction is well constructed. I recommend that authors add guideline references besides the AAPD to strengthen the background.

- “Therefore, the study aims to assess pacifier usage among a group of Saudi 2- to 4- year-old healthy children in Jeddah and to identify the relationship between pacifier usage and the child’s demographic data, maternal socioeconomic status, and knowledge of the harmful effect of dipping the pacifier in sweeteners.”

- It seems that the study explored more than demographic data, maternal socioeconomic status, and knowledge of the harmful effect of dipping the pacifier in sweeteners. Please, make the study aim more comprehensive.

- In the Methods or Discussion section, please provide explanation on the choice of the age range adopted (2-4 year-old children)

- The Discussion section is lacking regarding the study limitations. This is important to the readers to be aware of the main limitations that the study has. 

- The authors should further discuss and give further justifications on why first-born children seem more prone to use pacifiers.

Reviewer 3 Report

Comments and Suggestions for Authors

This is an important and highly relevant article, providing valuable insight into pacifier use among young children and its implications for oral health. The manuscript stands out as a reference for both dental and general medical fields, as it addresses an issue that intersects multiple specialties—including pediatric dentistry, orthodontics, periodontology, and general medicine. I recommend emphasizing the long-term periodontal effects of prolonged or sweetened pacifier use, not just the caries risk.

Please highlight that malocclusions from non-nutritive sucking habits can indirectly compromise periodontal health and oral hygiene. Note that sweetened pacifiers increase gingival inflammation, potentially leading to periodontal issues. The role of parents in maintaining children’s oral hygiene—even before age 4—should be stressed. I also suggest including references on the periodontal impact of pacifier use and on early oral hygiene practices. These additions would enhance the article’s multidisciplinary impact.

  1. Clinical Periodontal Context:
  2. Association between NNSH and Periodontal Risk:
  3. Sweetened Pacifiers and Periodontal Risk:
  4. Role of Parental Involvement in Oral Hygiene:
  5. Recommendation for Additional References

To further strengthen the manuscript, I suggest that the authors include references and discussions relating to the potential periodontal effects of pacifier use and sweetened pacifiers, as well as guidelines for parental involvement in early childhood oral hygiene. This would broaden the clinical relevance and reinforce the article’s importance across different dental and medical disciplines.

Reviewer 4 Report

Comments and Suggestions for Authors

Respected Authors,

I find your manuscript entitled "Pacifier Usage Among Saudi Children: A Cross-sectional Study in Jeddah, Saudi Arabia" interesting, and I consider that it can represent the starting point of more detailed research concerning the effects of using a pacifier on oral health status of children.

Please find below my observations and recommendations for the further improvement of the paper.

1. First of all, in my opinion, the manuscript is more suitable for another MDPI journal, such as Children, than for Healthcare. The study is well conducted, and the manuscript is well written, but it does not contain any results and conclusions concerning the impact of using a pacifier on oral health (such as dental malpositions, caries, gingivitis) or general health.

2. In the Introduction section, I would like to see a more complete presentation of the state of knowledge in this field, such as some concrete results of the studies in literature concerning the aspects presented in this section.

3. In the Methodology section, it is mentioned that the study was conducted in 2019. It is curious that the results are publicly presented after 6 years. Could you, please, explain this?

4. More details are necessary on the study instrument. You mentioned that a validated questionnaire was used in this study. Was this questionnaire validated in a previous study, or it was validated for the purpose of the present study? In case it was validated and used in previous studies, please mention references. 

5. Also regarding the questionnaire, please mention the number of questions, their type (closed, open), the average time needed to complete the questionnaire, and the Cronbach alpha value for internal consistency.

6. The association between the variable "attended a nursery" and pacifier use is unclear: in the Results section (lines 133-134) it is stated that "the use of pacifiers was higher 274 (62.1%) among those who did not attend a nursery compared to those who attended a nursery 167 (37.9%)". However, the results of regression analysis (lines 146-147) indicated an inverse relationship: "children who attended a nursery were significantly more likely to use pacifiers (AOR: 1.325 and 95% CI: 1.035-1.695)". Again, in the Discussion section (lines 198-199) it is mentioned that " children who did not attend a nursery had a higher chance of pacifier usage". Please clarify this.

Round 2

Reviewer 1 Report

Comments and Suggestions for Authors

Author has addressed all the comments by acknowledge the sample limitations, clearly explained the findings by addressing the possibility of gap between knowledge and behavior regarding avoiding pacifier. The last comment, abstract has been edited according this journal guideliness.

Overall the changes made by author, it seems the manuscript more comprehensive and readable.